# ON THE SURPRISING EFFICACY OF ONLINE SELF-IMPROVEMENT FOR EMBODIED MULTIMODAL FOUNDATION MODELS

## ABSTRACT

Foundation models trained on web-scale data have revolutionized robotics, but their application to low-level control remains largely limited to behavioral cloning. Drawing inspiration from the sample efficiency and success of reinforcement learning (RL) fine-tuning in large language models (LLMs), we propose a two-stage approach suited to robotics. The first stage, Supervised Fine-Tuning (SFT), fine-tunes pre-trained foundation models using goal-conditioned behavioral cloning and "steps-to-go" prediction objectives. In the second stage, this foundation enables the extraction of a well-shaped reward function and a success detector, eliminating the need for manual reward engineering and real-world instrumentation, and allowing robots to practice autonomously with minimal human supervision. Our experiments on both real-world and simulated robots demonstrate that the combination of SFT and online Self-Improvement is significantly more sample-efficient than supervised learning alone. Furthermore, the combination of our proposed approach with web-scale pre-trained foundation models enables rapid acquisition of new skills, allowing robots to generalize far beyond the behaviors observed in the imitation learning datasets used during training. These findings highlight the transformative potential of combining pre-trained foundation models with online fine-tuning to unlock new levels of autonomy and skill acquisition in robotics.

## 1 INTRODUCTION

Recent works have demonstrated that foundation models can be effectively fine-tuned to directly act as low-level robot policies (Brohan et al., 2023; Padalkar et al., 2023; Reed et al., 2022; Octo Model Team et al., 2024; Kim et al., 2024; Durante et al., 2024), and that they inherit significant generalization and robustness capabilities due to the web-scale pre-training of the foundation models from which they were derived. Such foundation agents present an exciting opportunity for the future of robotics, where a monolithic agent can plan, reason, and then execute actions in the environment. They also enable tighter transfer of methodologies between the adjacent fields of AI leveraging foundation models, such as Computer Vision and NLP. Throughout this work we will use the term "Multimodal Foundation Agent" (MFA) to refer to foundation models that act directly in an environment.

Thus far, the training regime for MFAs has largely been limited to behavioral cloning (i.e. supervised learning) (Brohan et al., 2023; Padalkar et al., 2023; Reed et al., 2022; Octo Model Team et al., 2024; Kim et al., 2024; Durante et al., 2024). In contrast, from the literature on Large Language Models (LLMs) we observe that after the initial pre-training, post-training for downstream tasks is typically divided into two stages: 1) Supervised Fine-Tuning (SFT), followed by 2) Reinforcement Learning (RL) where models improve their performance on downstream tasks such as math, coding, as well as aligning with human preferences (RLHF) (Ouyang et al., 2022). RL-Tuning of LLMs has been shown to markedly, and rapidly, improve downstream task performance beyond the SFT stage (Stiennon et al., 2020; Ouyang et al., 2022), and has become a critical stage in the training recipe of foundation models (Achiam et al., 2023; Team et al., 2024; Dubey et al., 2024).

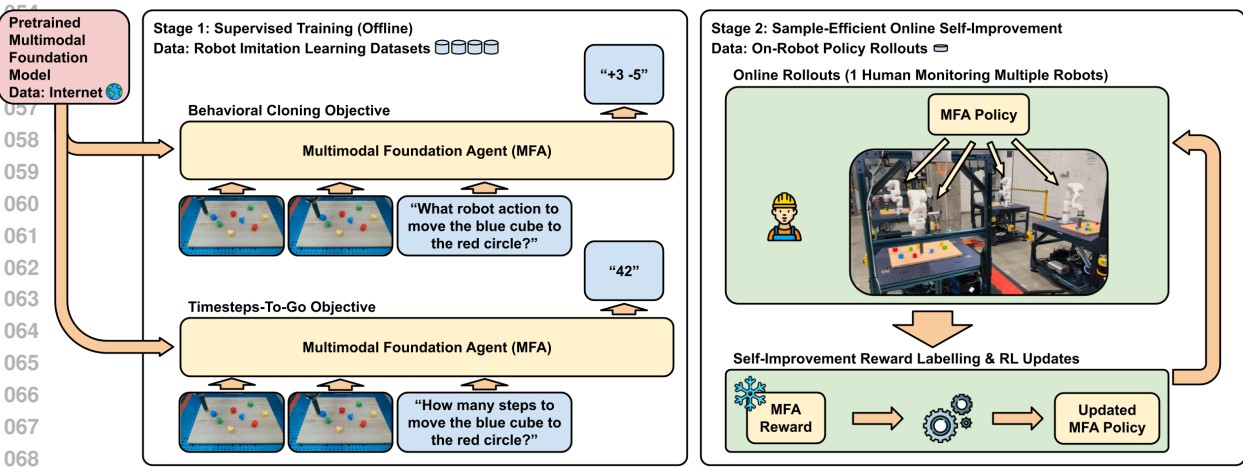

Figure 1: **Overview of our proposed two-stage fine-tuning approach.** Stage 2 online self-improvement efficiently improves robot policies and enables learning novel out-of-distribution tasks.

Despite the unique algorithmic and engineering challenges of investigating RL-tuning for MFAs in the context of robotics, the aforementioned sample-efficiency and performance gains from the LLM literature strongly motivate its investigation. In this work we directly tackle these challenges and design a two-stage framework inspired by LLM post-training processes: In Stage 1 "Supervised Fine-Tuning" (SFT), given a goal-conditioned imitation learning dataset we fine-tune MFAs using two objectives: 1) Behavioral Cloning, and 2) Predicting the number of "steps-to-go" to accomplish desired goals. In Stage 2 "Online Self-Improvement", we leverage the model's own steps-to-go predictions to derive an effective reward function and success detector, enabling 1 human operator to monitor multiple robots as they practice downstream tasks. Critically, our data-driven reward design circumvents the need for ground-truth rewards, and leverages the robustness and generalization properties of the underlying foundation models.

Through extensive experiments on two robot embodiments, LanguageTable (Lynch et al., 2023) and Aloha (Zhao et al., 2023; Aldaco et al., 2024), in the real-world and simulations, we demonstrate the surprising efficacy of our proposed fine-tuning framework. Our results demonstrate that Stage 2 fine-tuning very sample-efficiently and robustly improves policy performance. Furthermore, we highlight that it is more efficient to distribute robot time budget between imitation data collection for Stage 1 and Stage 2 self-improvement, rather than allocating the full robot time for Stage 1 data collection alone. We then demonstrate the immense value of the webscale pre-training of foundation models. Pre-taining not only results in significant sample-efficiency, but also unlocks the ability for robots to autonomously practice and acquire new skills generalizing far outside the distribution of tasks seen during Stage 1.

Our work highlights the transformative potential of combining pre-trained foundation models with online fine-tuning to unlock new levels of autonomy and skill acquisition in robotics. In Section 7 we discuss a series of open questions that would engender fruitful research endeavours for future work. With the proliferation of open-source multimodal foundation models (Beyer et al., 2024; Dubey et al., 2024; Liu et al., 2024; Wang et al., 2024), and hardware efficient fine-tuning methods (Hu et al., 2021; Dettmers et al., 2024), we believe such research agendas could be effectively studied by a broad community of robotics researchers. Our anonymous supplementary videos website can be found at: `https://sites.google.com/view/mfa-self-improvement/home`

## 2 BACKGROUND

**PaLI Vision-Language Foundation Model** While our investigations in this work are independent of the choice of underlying multimodal foundation model used, throughout this work we use the 3 billion parameter PaLI-3B (Chen et al., 2022; 2023) vision-language model as the base pretrained foundation model that we will be fine-tuning for robotics tasks. A PaLI model receives as input one

or more images alongside text, and provides text as output. At a high level, the PaLI architecture is comprised of two components: 1) a Vision Transformer (ViT) (Parmar et al., 2018), and 2) an encoder-decoder Transformer (Vaswani et al., 2017). Input images are processed by the ViT into a sequence of "visual tokens". The sequence of visual tokens is concatenated with the tokenized text input and fed into the Transformer encoder, and the Transformer decoder outputs text tokens. The PaLI architecture is initialized from a Transformer encoder-decoder model (language) and ViT (vision) that are pretrained separately in a unimodal fashion. The model is subsequently trained jointly with a variety of vision-language training objectives to obtain a multimodal foundation model. For further details regarding PaLI model, we refer the interested reader to (Chen et al., 2022; 2023). We emphasize that our framework is independent of the choice of underlying multimodal foundation model used.

**RT-2** Brohan et al. (2023) introduce a model family, dubbed RT-2, that enables vision-language foundation models (VLMs) to directly perform closed-loop robot control. The two VLMs considered in that work are PaLI (Chen et al., 2022; 2023) and PaLM-E (Driess et al., 2023), both of which take images alongside text as input, and provide output in the form of text tokens. To enable these VLMs to act as robot policies, continuous robot actions are discretized and mapped onto the linguistic token space. Given image and text inputs, the VLMs are fine-tuned via behavioral cloning (BC, i.e. supervised learning) to predict the tokenized robot actions. While the methods we present in this work are independent of the choice of underlying model and architecture, throughout this work our robot policy architectures are equivalent to RT-2 using the PaLI VLM.

## 3 METHODOLOGY

Given access to a goal-conditioned behavioral cloning dataset, our focus in this work is to design an effective and sample-efficient procedure for fine-tuning pretrained multimodal foundation models in order to obtain performant robotic MFAs. Our proposed fine-tuning framework is composed of two stages: 1) Supervised Fine-Tuning (SFT) wherein we train MFAs using goal-conditioned behavioral cloning as well as "steps-to-go" prediction objectives, and 2) Online Self-Improvement (Online RL) wherein MFA policies autonomously practice downstream tasks and rapidly improve themselves via self-predicted rewards.

A critical challenge of reinforcement learning for robotics, and in particular for manipulation tasks, is the problem of reward engineering. Designing effective reward functions requires repeated trial-and-error iterations of training RL policies and patching reward definitions to arrive at intended outcomes. Furthermore, even with a perfect reward function, significant research and engineering effort must be dedicated to measuring rewards in the real-world. Thus, manual reward design is untenable as we move towards a future where we train robots to accomplish increasingly broad sets of tasks. A key feature of our proposed approach is that it overcomes this obstacle via learning data-driven reward functions that also inherit robustness and generalization properties from the web-scale pre-training of the foundation models used to build the MFAs.

### 3.1 STAGE 1: SUPERVISED FINE-TUNING (SFT)

The first stage of our frameworks consists of an offline Supervised Fine-Tuning (SFT) stage. We assume access to a goal-conditioned imitation learning dataset $\mathcal{D}$ consisting of a collection of episodes $\tau = \{(o_t, a_t, g_\tau)\}_{t=0}^T$, where $o_t$ and $a_t$ denote observation and action at timestep $t$ respectively, and $g_\tau$ denotes the goal for episode $\tau$. We assume that all trajectories in the dataset end in a state where the episode goal is accomplished. In the case of single-task datasets, we treat them as a goal-conditioned dataset where all episodes share the same goal. Given a dataset $\mathcal{D}$ and pretrained multimodal foundation model, we instantiate the MFA (e.g. RT-2 (Brohan et al., 2023) parameterization), and fine-tune the model using the following supervised learning objectives:

$$\mathcal{L}_{\text{BC}}(\text{MFA}) = -\mathbb{E}_{(o_t, a_t, g_\tau) \sim \mathcal{D}} \Big[ \log p_{\text{MFA}}\big(a_t \mid o_t, \text{Question}_{\text{action}}(g_\tau)\big) \Big]$$

$$\mathcal{L}_{\text{steps\_to\_go}}(\text{MFA}) = -\mathbb{E}_{(o_t, a_t, g_\tau) \sim \mathcal{D}} \Big[ \log p_{\text{MFA}}\big(\text{length}(\tau) - t \mid o_t, \text{Question}_{\text{steps\_to\_go}}(g_\tau)\big) \Big]$$

$\mathcal{L}_{\text{BC}}$ denotes goal conditioned behavioral cloning loss, where we maximize the likelihood of a dataset action conditioned on the observation and a text sequence $\text{Question}_{\text{action}}(g_\tau)$ representing

---

**Algorithm 1:** Stage 2 Self-Improvement Loop

---

**Input:** Policy model and frozen reward computation model taken from Stage 1 checkpoints

**while** *true* **do**

    Using the current policy collect enough robot rollouts for $N$ update steps with batch size $B$;

    **for** *each rollout* **do**

        Compute Monte Carlo returns using Equation 2: $R_t \leftarrow \sum_{i=t}^{T} \gamma^{i-t} \cdot r(o_t, a_t, o_{t+1}, g)$;

        Place $(o_t, a_t, g, R_t)$ tuples in the replay buffer;

    **end**

    Shuffle the buffer, update with REINFORCE $[-c \cdot R_t \cdot \log p_{\text{MFA}}(a_t|o_t, \text{Question}_{\text{action}}(g))]$;

    Empty the replay buffer if there are any remaining elements;

**end**

---

the desired goal. In this work we use $\text{Question}_{\text{action}}(g_\tau) =$ `"What robot action to $g_\tau$?"`. The objective $\mathcal{L}_{\text{steps\_to\_go}}$ teaches the MFA to predict how many environment timesteps away the robot is from accomplishing an intended goal. In this work we use $\text{Question}_{\text{steps\_to\_go}}(g_\tau) =$ `"How many steps to $g_\tau$?"`, and in 3.2 we will observe the critical role of this objective.

Depending on the domain, at this stage we can include additional auxiliary supervised objectives. As an example, in our experiments with the LanguageTable domain, conditioned on the first and last image of an episode we ask the model to predict what instruction was performed in that episode.

### 3.2 STAGE 2: ONLINE SELF-IMPROVEMENT (ONLINE RL)

In Stage 2, our goal is to fine-tune the MFA with online RL, with the hopes that it will lead to rapid and significant performance improvements on desired downstream tasks. As we will see later on in our experiments, downstream tasks may even be significantly different than those that appeared in the dataset $\mathcal{D}$ used for Stage 1 training (Sections 5.3.1 and 5.3.2).

**Reward Function Definition** Let,

$$d(o, g) := \mathbb{E}_{p_{\text{MFA}}\left(\text{steps\_to\_go}|o, \text{Question}_{\text{steps\_to\_go}}(g)\right)} \left[\text{steps\_to\_go}\right] \tag{1}$$

denote the expected value of "steps to go" in order to accomplish goal $g$ given observation $o$, as predicted by the MFA model obtained after Stage 1. The reward function we use for online RL training is defined as follows,

$$r(o_t, a_t, o_{t+1}, g) := d(o_t, g) - d(o_{t+1}, g) \tag{2}$$

Intuitively, this reward function predicts how much closer the robot got towards accomplishing goal $g$ after taking action $a_t$. As the reward function is derived from $d(o, g)$, which is a function of the MFA itself, we refer to our online RL fine-tuning process as "Self-Improvement". The choice of using the expected value in equation 1 is for simplicity and alignment with the notion of a value function in RL. We leave investigations of alternate definitions such as CVaR (Alexander & Baptista, 2004) for risk-aware policies, or distributional RL (Bellemare et al., 2023), to future work.

**Self-Improvement Procedure** To perform Stage 2 fine-tuning, we take a frozen Stage 1 checkpoint to use for reward function calculations, and initialize the Stage 2 policy from a Stage 1 checkpoint as well. The checkpoints for the reward and policy models are not necessarily identical as the best validation losses for $\mathcal{L}_{\text{BC}}$ and $\mathcal{L}_{\text{steps\_to\_go}}$ can happen at different points over the course of Stage 1 training. Within one iteration of our Stage 2 Self-Improvement loop, using the current policy we collect enough robot trajectories to perform $N$ model update steps. Subsequently, for each trajectory, per timestep, we compute the Monte Carlo returns $R_t \leftarrow \sum_{i=t}^{T} \gamma^{i-t} \cdot r(o_t, a_t, o_{t+1}, g)$ and place elements $(o_t, a_t, g, R_t)$ in a shuffled replay buffer. We then perform $N$ policy updates using the REINFORCE loss $\left[-c \cdot R_t \cdot \log p_{\text{MFA}}(a_t|o_t, \text{Question}_{\text{action}}(g))\right]$. The replay buffer is then cleared out and the next iteration begins. Algorithm 1 above presents our Stage 2 Self-Improvement procedure. In simulation experiments we found that using a small positive multiplicative factor $c$ in the REINFORCE loss plays a significant role in ensuring the model trains stably. Throughout this

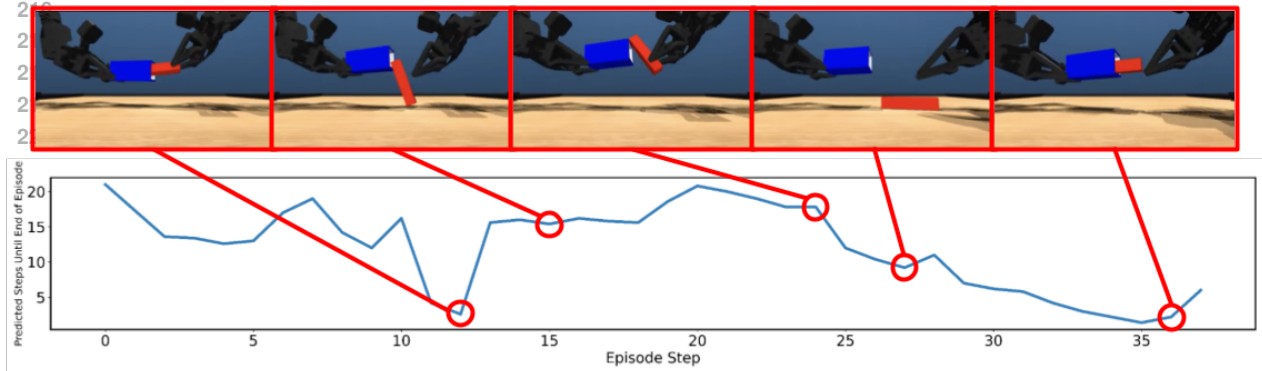

Figure 2: An example trajectory from the Aloha Single Insertion Task and a plot of model predictions for steps-to-go, $d(o, g)$. Key moments: 1) Model believes episode is about to complete successfully, 2) Policy accidentally drops the peg and $d(o, g)$ increases, 3) Policy regrasps the peg from a bad angle not suitable for insertion so $d(o, g)$ remains high, 4) Policy drops the peg, enabling regrasping correctly which reduces $d(o, g)$, 5) Policy is pushing the peg inside and $d(o, g)$ marks that episode is about to succeed.

work we used $c = \texttt{5e-2}$. Despite our goal of sample-efficient RL, we choose to perform on-policy RL without data reuse due to the stability of on-policy RL methods (Van Hasselt et al., 2018), and leave the investigation of off-policy RL methods for future work.

**Success Detection**   We find that it is important for robot episodes to terminate upon successfully reaching the intended goal state. Otherwise, a significant portion of the collected data will include the robot being in a successful terminal state. In settings where we do not have a ground-truth success detector, as in our real-world experiments, we use the following success indicator derived from the frozen reward model checkpoint: $\text{success}(o, g) := \mathbb{1}[d(o, g) \leq s]$, with $s$ being a very small number of timesteps. We found this formulation of success detection to be very robust even in low data regimes, and significantly more reliable than explicitly including a success detection binary classification objective in Stage 1. Throughout our work we use $s = 3$ unless noted otherwise.

## 4   Intuition on Reward Function

**Mathematical Intuition**   For the interested reader, in Appendix E we discuss how our proposed Stage 2 procedure *leads to policies that more efficiently achieve intended goals while being implicitly regularized to stay close to the dataset policy $\mu$!* We also highlight a supplementary python notebook implementing our two stage fine-tuning procedure on a pointmass goal-reaching domain.

**Visual Intuition For Our Choice of Reward Function**   We can also attempt to build our intuition regarding the efficacy of steps-to-go prediction via visualizing model predictions on domains of interest. Figure 5 visualizes an example trajectory on the Aloha Single Insertion Task (task details provided in 5.1). The caption in Figure 5 walks the reader through the level of intricate details that the MFA model is able to learn. We provide additional visualizations – including on the LanguageTable domain – in the form of videos on our anonymous supplementary content website.

## 5   Experiments

In our experiments we seek to validate our proposed self-improvement framework and answer the following five questions:

- **Q1:** Does our self-improvement procedure improve performance on downstream tasks beyond the supervised learning stage?
- **Q2:** Is our self-improvement procedure, which depends on RL, reliable and reproducible enough to be employed for real-world robotics?
- **Q3:** Is the combination of supervised learning and self-improvement a more efficient procedure for obtaining performant policies, compared to supervised learning alone?

- **Q4:** What is the contribution of the web-scale pretraining of multimodal foundation model?
- **Q5:** Can we leverage the pretraining knowledge embedded into the MFA to push generalization abilities, and perform Stage 2 Self-Improvement on tasks that generalize beyond what was seen in the imitation dataset?

We study these questions using the LanguageTable (Lynch et al., 2023) and Aloha (Zhao et al., 2023; Aldaco et al., 2024) robot embodiments, with experiments in both simulation and the real-world (please refer to Appendix B for details regarding the robotic domains used). As mentioned in Section 2, throughout this work we will use the PaLI (Chen et al., 2022; 2023) vision-language model as our base pretrained foundation model. The inputs to our PaLI MFA are always two images and a text sequence, and the outputs are a sequence of tokens. To employ PaLI models as policies, we follow the RT-2 (Brohan et al., 2023) policy parameterization and predict tokenized actions. Thus, our Stage 1 behavioral cloning policies are exactly equivalent RT-2 policies and will serve as key baselines. To use the PaLI model for predicting steps-to-go, we also map the range of integers $[0, T]$ onto the PaLI model's output token space. We refer the interested reader to Appendix A for details regarding tokenization. In Stage 1 we do not freeze any parameters in the model, and fine-tune both the Transformer and the ViT backbone. In Stage 2 we do not further fine-tune the ViT portion of the model. This was an early decision in our project in hopes of improved stability, and we did not ablate this choice.

## 5.1 SELF-IMPROVEMENT IS ROBUST, EFFECTIVE, AND MORE EFFICIENT THAN SUPERVISED LEARNING ALONE

### 5.1.1 SIMULATED LANGUAGETABLE

The dataset we use to train Stage 1 policies for the simulated LanguageTable domain is the one provided by the original work (Lynch et al., 2023). This dataset consists of 181,020 human-generated trajectories, with 78,623 unique instructions describing the goals of the trajectories. We subsample this dataset to create 3 new datasets 10%, 20%, and 80% of the original size. For each dataset size we take the following procedure: First, we perform the Stage 1 supervised fine-tuning of the PaLI MFA. We use the checkpoint at the best imitation validation loss as the supervised policy checkpoint, and the one at the best steps-to-go prediction validation loss for reward computation. We perform Stage 2 fine-tuning with 3 seeds to validate the reliability of the self-improvement procedure. While the LanguageTable dataset contains a variety of tasks, we perform Stage 2 fine-tuning on the Block2Block tasks, e.g. `"move the blue moon to the red pentagon"`. We stop Stage 2 training when policy success rates appear to plateau.

**Results** The first plot in Figure 3 presents our results on the simulated LanguageTable domain, where orange markers represent policy performance after Stage 1, and blue markers represent policy performance after Stage 2. As can be observed, across all dataset sizes (10%, 20%, 80%), our proposed self-improvement procedure leads to very significant improvement in success rates (minimum 1.5x performance boost), with incredible sample-efficiency in terms of number of episodes (less than 2% extra episodes collected in Stage 2). As an example, by training a 10% data Stage 1 policy with 1% additional episodes in Stage 2, we obtain policies that outperform both the 20% and 80% data Stage 1 policies. Furthermore, as evidenced by Figure 8 left (Appendix D), across random seeds our Stage 2 process is stable and reproducible, with the individual blue markers representing individual experiments tightly packed together.

### 5.1.2 REAL-WORLD LANGUAGETABLE

The significant sample-efficiency and robustness of our results suggest that our self-improvement procedure may indeed be applicable for real-world robotics. To this end, we apply our two-stage fine-tuning framework to the real-world LanguageTable domain, in two settings of using 20% and 80% of the real-world LanguageTable dataset (Lynch et al., 2023). As in the simulated setting, we apply our Stage 2 process on the Block2Block subset of tasks. Experiments are run for approximately 20 hours each, with 1 human operator monitoring and periodically resetting 3-4 LanguageTable robot stations simultaneously. For details on the real-world LanguageTable experimentation protocol we refer the interested reader to Appendix C. We run the 80% data experiment once using 3 robot stations, and run the 20% data experiment twice, once with 3 and once with 4 robot stations. As described in Section 3.2, success detection for episode termination is performed automatically by our system, and the sole responsibility of the human operator is to monitor the robots and periodically reset the blocks on the stations.

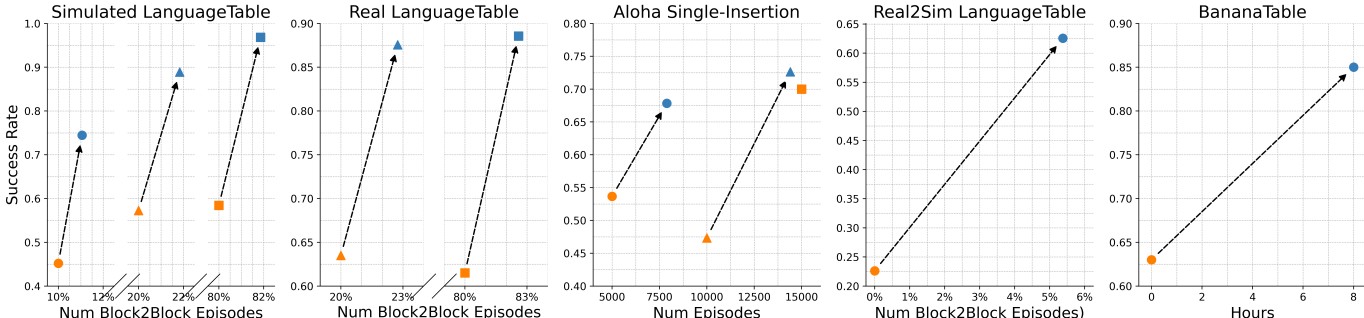

Figure 3: **Stage 2 Self-Improvement Results.** Orange: Stage 1 (equivalent to RT-2 Brohan et al. (2023) baseline). Blue: Stage 2 Self-Improvement. Our results in simulated and real LanguageTable, and well as Aloha domain, demonstrate that our proposed two-stage approach achieves higher success rates significantly more sample-efficiently than supervised learning alone. Our Real2Sim LanguageTable and in particular BananaTable results demonstrate that the combination of Stage 2 and web-scale pre-training enables policies to acquire novel skills far outside the Stage 1 imitation learning dataset. Variations across random seeds are small, highlighting the robustness of our approach. Values above are averaged across 3 seeds (unaggregated results in Figure 7, Figure 8). While Stage 1 LanguageTable datasets contain varied tasks, for fairness, the x-axes in the plots above count number of Block2Block episodes (normalized by number of Block2Block episodes in full dataset).

**Results** Figure 3 presents our results. As can be seen, for both the 20% and 80% data settings, our Stage 2 self-improvement procedure improves policy success rate from ∼60% up to ∼80%-85%, all within ∼3% additional Block2Block episodes. To put this into perspective, this means that with a total amount of experience equivalent to ∼23% (Stage 1 + Stage 2), we obtain policies that far exceed the Stage 1 BC policies (i.e. RT-2) that used 80% of the real-world LanguageTable dataset. Furthermore, as opposed to the 1-to-1 human-to-robot ratio during imitation learning data collection for Stage 1, the Stage 2 process requires only $\frac{1}{4}$ of the human effort due to the 1-to-many human-to-robot ratio enabled by our proposed approach.

### 5.1.3 SIMULATED ALOHA SINGLE INSERTION TASK

We also validate our proposed fine-tuning framework on a second robot embodiment, the bimanual Aloha manipulation platform (Zhao et al., 2023; Aldaco et al., 2024). We designed and collected data for a bimanual insertion task, where the left gripper must pick up a socket, and the right gripper must pick up a peg and insert that peg into the socket. Figure 6 presents a visualization of this task, with videos available on our supplementary materials website. Due to the single-task nature, much smaller imitation datasets, much more complex observations, and 70-dim action space, this presents a challenging setting for further validation of our proposed process. For details on the task and how the datasets were created, we refer to Appendix B.3. We create 3 imitation dataset sizes of 5K, 10K, and 15K trajectories. We apply our two-stage process on 5K and 10K dataset sizes, and report results for supervised learning on the 15K dataset as well to better situate the numbers. The only differences in methodology compared to LanguageTable domain are the following: 1) To initialize the Stage 2 policy checkpoint we do not take the best validation checkpoint, as we saw that further training the supervised policy lead to much more improved performance. 2) Since the exact success state is difficult to observe from the robot camera observations, we add a small positive constant to the reward function when the robot reaches a successful state. Our task and collected data will be open-sourced in an upcoming contribution to the Aloha simulation repository (Aldaco et al., 2024).

**Results** Figure 3, middle, presents our results. As can be seen, policies trained with 5K+2.5K episodes (Stage 1 + Stage 2) outperform policies trained with 10K imitation episodes (Stage 1 only, RT-2), and rival the success rate of those trained with 15K supervised episodes (Stage 1 only, RT-2).

> **A1, A2:** Our proposed Stage 2 fine-tuning procedure significantly improves policy performance on downstream tasks, is reliably reproducible across experiment seeds, and is robust enough to be strongly effective on real-world robot training.
> **A3:** Within a given budget of robot episodes, we can obtain more performant robot policies by distributing the budget between our proposed Stage 1 and Stage 2 fine-tuning stages, as opposed to allocating that budget purely for Stage 1 imitation data collection.

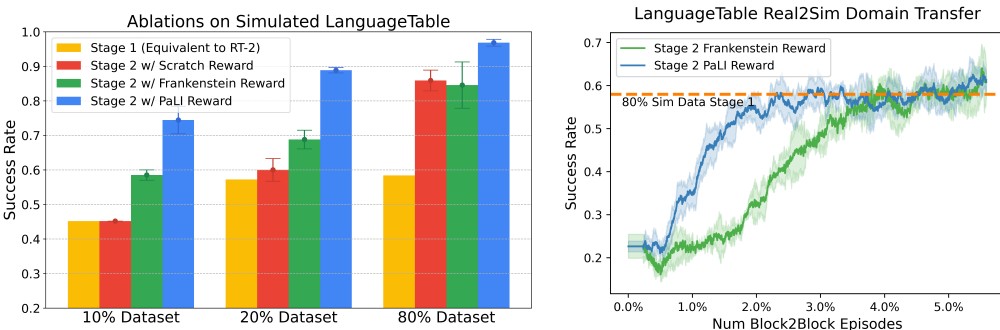

Figure 4: **Left** Our ablation results demonstrate the critical role of the web-scale pre-training of foundation models for enabling effective Stage 2 training, in particular in the small dataset size regime. **Right** LanguageTable Real2Sim domain transfer results.

## 5.2 Importance of Foundation Model Pretraining

It is critical to study to what extent the significant benefits of our proposed self-improvement procedure are afforded by the webscale pretraining of the PaLI (Chen et al., 2022; 2023) foundation model we start from. As described in Section 2, the PaLI model is initialized from a pretrained ViT model (trained unimodally using vision tasks) and a pretrained language Transformer model (trained unimodally using language tasks), which are connected to form the PaLI architecture, and subsequently co-trained on multimodal vision-language tasks. To ablate the effect of the multimodal knowledge embedded into PaLI, we can run our proposed two-stage fine-tuning process starting from alternative variations of the PaLI model:

- **Scratch:** where we use the PaLI architecture but with randomly initialized parameters.
- **Frankenstein:** where we take the version of the PaLI model that connects the pretrained ViT model to the pretrained language Transformer, but without the PaLI vision-language co-training. We refer to this model as the "Frankenstein" model, referencing how the ViT and the Transformer are "Frankensteined together".

Similar to Section 5.1.1, we compare these variations on the Simulated LanguageTable domain, using the 10%, 20%, and 80% dataset sizes, performing Stage 2 fine-tuning on the Block2Block subset of tasks. Each experiment is ran with 3 random seeds. Despite our best efforts and very long training runs, we observed that Stage 1 supervised policies derived from Scratch or Frankenstein variations very significantly underperformed PaLI Stage 1 policies. Hence, we focus our ablation on the Stage 2 self-improvement process, where the policy is initialized from the PaLI Stage 1 checkpoints, and the reward model uses Scratch or Frankenstein checkpoints. Figure 4 Left presents our results. There is a clear ordering in performance, where the PaLI is best, followed by Frankenstein, and then Scratch reward model. The Scratch reward models leads to high variance results across random seeds, and struggles to provide any meaningful improvements in the low-data regimes, to the point that in the 10% data regime Stage 2 fine-tuning could not improve the Stage 1 policy. While better than Scratch, Stage 2 with Frankenstein reward models is also significantly worse than using PaLI reward models. In fact, Stage 2 fine-tuning with PaLI reward models in the 20% regime leads to better policies than Stage 2 fine-tuning with Frankenstein reward models in the 80% regime! These results clearly demonstrate the immense value that webs-cale multimodal pre-training brings to our self-improvement procedure.

> **A4:** Foundation model pre-training leads to significantly better Stage 2 policies, and is a key enabler of sample-efficiency.

## 5.3 Generalization

A capability unlocked by the combination of our proposed self-improvement process and the use of pretrained multimodal foundation models is that, during Stage 2 self-improvement policies can

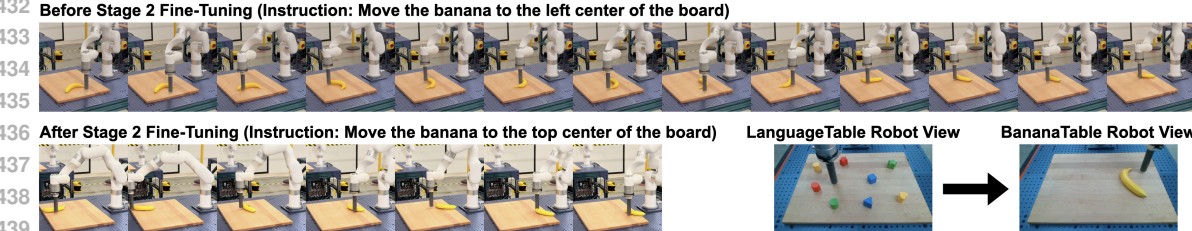

Figure 5: **Strong Generalization to BananaTable. Top** Before Stage 2 fine-tuning on the BananaTable domain, the policy struggles to effectively maneuver a banana across the table due to the difficult geometry. **Bottom Left** After Stage 2 fine-tuning policies are visibly more proficient at the BananaTable task (videos on our supplementary website). **Bottom Right** Prior to Stage 2 BananaTable fine-tuning, the policy and reward models have never seen the BananaTable task, creating a very challenging generalization problem.

practice novel tasks that were not covered by the imitation learning dataset. In this section we present results for two increasingly difficult forms of generalization.

### 5.3.1 DOMAIN TRANSFER BETWEEN SIMULATION AND REAL

In this section we investigate domain transfer between simulation and real. Sim2Real is an important class of approaches for robotics with many successes (Pinto et al., 2017; Tan et al., 2018; Akkaya et al., 2019; Rao et al., 2020; Kataoka et al., 2023), and can significantly reduce the amount of real-world experience needed to train performant robot policies. To make experimentation simpler, in this section we investigate the inverse problem of Real2Sim transfer on the LanguageTable domain. We train Stage 1 models using 80% of the *real-world* LanguageTable dataset, and perform Stage 2 self-improvement in the *simulated* LanguageTable environment. Similar to our ablation in section 5.2, we also train Stage 2 models using the "Frankenstein" reward model variant to highlight the role of foundation model pretraining in enabling domain transfer.

Figure 4 right, presents our results. As can be seen, with a mere number of episodes equivalent to 3% of Block2Block episodes in the simulated LanguageTable dataset, our Stage 2 self-improvement procedure improves policy performance from $\sim 22\%$ to $\sim 59\%$. This performance is equivalent to PaLI Stage 1 models (i.e. RT-2 behavioral cloning) trained with 80% of the simulated LanguageTable dataset. Additionally, Figure 4 right demonstrates that the Frankenstein model leads to a significantly slower self-improvement procedure, highlighting the key role of PaLI pre-training. Given our strong real-world LanguageTable results in section 5.1.2, we expect our Real2Sim results to be strongly indicative of Sim2Real transfer as well.

### 5.3.2 STRONG GENERALIZATION TO LEARNING NOVEL SKILLS

Lastly, we test the generalization ability of our self-improvement approach via an experiment we dub "BananaTable" (Figure 6, top right). Starting from a real-world LanguageTable policy that was Stage 2 fine-tuned for the Block2Block tasks and the corresponding reward model (Section 5.1.2), we perform futher Stage 2 fine-tuning but for the BananaTable task, where we replace the LanguageTable blocks with a single prosthetic banana and request policies to push the banana to various locations on the board.

Prior to this experiment, the policy and reward function have never seen a banana or the table without blocks. Thus we are solely relying on the generalization abilities of the PaLI model underneath. Not only is the BananaTable scene visually different from LanguageTable, requiring semantic generalization, but manipulating bananas effectively necessitates learning new skills compared to the ones used for manipulating LanguageTable blocks, requiring behavioral generalization. As an example, due to its geometry, inaccurate pushing of a banana results in it rotating around itself instead of moving in the intended direction. The videos in our supplementary website demonstrate that within $\sim 8$ hours of training using 2 robot stations, the policy becomes visibly more proficient at accomplishing the BananaTable tasks ($\sim 63\% \longrightarrow \sim 85\%$ success rate, Figure 3).

> **A5:** Unlike prior work such as RT-2 (Brohan et al., 2023) where foundation models have demonstrated semantic generalization (e.g. executing the same pick and place motion but in a different context), our proposed self-improvement procedure enables policies to not only refine their behaviors during domain transfer (Real2Sim, Section 5.3.1), but also rapidly acquire new skills that are strongly beyond the distribution covered by the imitation learning datasets they are provided (BananaTable, Section 5.3.2).

## 6 RELATED WORKS

**Pretrained Models As Robot Policies**  A number of works leverage pre-trained multimodal foundation models to obtain performant robot policies. RT-2 Brohan et al. (2023) fine-tunes variants of the PaLI (Chen et al., 2023) vision-language model using behavioral cloning, demonstrating strong performance and generalization gains from the use of pre-trained models. This approach was further validated by applying to the Open X Embodiment (Collaboration et al., 2023) dataset containing over 1M robot trajectories from 21 institutions (e.g. Octo (Octo Model Team et al., 2024) and OpenVLA (Kim et al., 2024)). Shah et al. (2023) train a foundation model for visual navigation, demonstrating that it can be leveraged to control various embodiments, and sample-efficiently fine-tuned to adapt to new observation modalities, as well as new navigation tasks and domains.

**Improving Robot Policies Without Ground-Truth Rewards**  A significant challenge of improving robot policies is that commonly we do not have access to ground-truth reward functions, whether due to the challenge of designing one, or difficulty in measuring them. An important class of works (Bhateja et al., 2023; Ma et al., 2022; Sermanet et al., 2018) learn latent observation representations on top of which rewards and value functions can be defined. In contrast, our approach of predicting timesteps until end of episodes directly leverages the existing input-output space and loss functions of existing large pre-trained models, making it much simpler to implement. A number of prior works also leverage time distances between states to learn policies. Hartikainen et al. (2019) use unsupervised interactions to learn distances between states, while also using imitation datasets to guide policies towards goals. Predicted distances to goals are used as negative rewards and for goal-conditioned RL. In an offline setting, Hejna et al. (2023) model the distribution of timesteps between states in imitation learning datasets and approximate shortest paths. Policies are extracted by weighing actions by their reduction in distance estimates. Many alternative approaches to robot learning without rewards have been explored as well. Kumar et al. (2022) use a heuristic of labeling the last $n$ trajectories of an episode with $+1$ rewards and the rest with 0. They demonstrate that offline and online RL, using the combination of target task and pre-existing data, can be used for sample-efficiently improving robot policy performance. (Eysenbach et al., 2022) demonstrate that a particular form of contrastive learning corresponds to a form of goal-conditioned Q-Learning. (Chebotar et al., 2021) demonstrate the offline goal-conditioned RL with relabeled goals can lead to sample-efficient downstream fine-tuning for new tasks. RobotCat (Bousmalis et al., 2023) trains a large behavioral cloning Transformer with a similar architecture as Gato (Reed et al., 2022) on a diverse set of robotics tasks. They demonstrate that policy performance can be improved by rolling out the policies, relabeling episodes with accomplished goals, and adding the trajectories back into the imitation dataset. However, it is important to note that using hindsight relabeled supervised learning as a policy improvement procedure can have important failure cases (Ghugare et al., 2024).

## 7 FUTURE WORK AND LIMITATIONS

Our work has clearly demonstrated the immense potential of the combination of pre-trained multimodal foundation models and online self-improvement towards efficiently obtaining performant robot policies that also exhibit strong generalization capacities. There still exist, however, many important avenues for future work: 1) Our approach uses on-policy REINFORCE for simplicity which does not reuse any collected data in Stage 2. Off-policy methods have the potential to even more substantially improve Stage 2 sample-efficiency. 2) Training large models requires significant compute budgets. Understanding whether our framework is amenable to parameter-efficient tuning methods (e.g. LoRA (Hu et al., 2021)) is critical towards enabling broad access to Stage 2 fine-tuning. 3) What are the failure cases of our reward formulation? Anecdotally, we have seen that if we continue training Stage 2 for past the peak performance, the success rate of the policies can begin to degrade. Is this due to gaps between our reward formulation and intended task outcome? We hope that the strong results presented in this work motivate the broader research community to investigate these fruitful avenues of future research.

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

# A    TOKENIZATION

## A.1    LANGUAGETABLE

**Actions**    We represnt LanguageTable actions via 4 tokens: $+/-$, token representing 0-10, $+/-$, token representing 0-10. The continuous 2D actions are binned to fall into this representation.

**Timesteps**    We represent timesteps until end of episode using one token, which represents a number between 0-50.

## A.2    ALOHA

**Actions**    As discussed in B.3 our action space is $5 \times 14$ dimensions. We represent each dimension as 1 token, meaning the model outputs 70 tokens. Each token represents a number from 0-255. The continuous Aloha actions are discretized and binned into these 256 bins.

**Timesteps**    We represent timesteps until end of episode using one token, which represents a number between 0-300.

**Joints**    As input we provide the model with the current joint positions, i.e. we append 14 tokens to the input, where each token represents a number from 0-255. The continuous Aloha joints are discretized and binned into these 256 bins.

# B    ENVIRONMENTS AND TASKS

Figure 6 presents a visualization of the tasks used in this work.

## B.1    LANGUAGETABLE

The LanguageTable domain (Lynch et al., 2023) has a 2D action space representing delta movement in the x-y plane. The tasks we perform Stage 2 on are the Block2Block subset of tasks which contain instructions of the form ``move the blue cube to the green star". The datasets used in Stage 1 are those provided by the original paper. The two images given to PaLI represent the current and previous frame as viewed by the LanguageTable robot camera (Figure 6, top left).

## B.2    BANANATABLE

In the BananaTable task we remove all blocks from the LanguageTable stations and replace them with a single banana. The instructions for the BananaTable task have the form, "X the banana to the Y of the table.", where X is a set of verbs synonomous with pushing, and Y is one of left, top left, top center, top right, right, bottom right, bottom, bottom left, center.

## B.3    ALOHA

The Aloha domain is 14 degree of freedom joint-space controlled robot. As opposed to the default 50Hz, we operate the environment at 10Hz. A common design choice in the Aloha domain (Zhao et al., 2023) is to train policies to predict $N$ actions into the future. We use $N = 5$ which results in an action space that is 70-dimensional ($14 \times 5$).

The Aloha environment has 4 cameras. To turn them into two images to pass to our PaLI models, we stack two images into one image with a black buffer in between. Figure 6 bottom left and bottom right show an example of the two images given to PaLI.

We designed and collected data for a bimanual insertion task, where the left gripper must pick up a socket, and the right gripper must pick up a peg and insert that peg into the socket. We collected 800 demonstrations using a VR headset to view the Mujoco simulation, and using the real-world Aloha

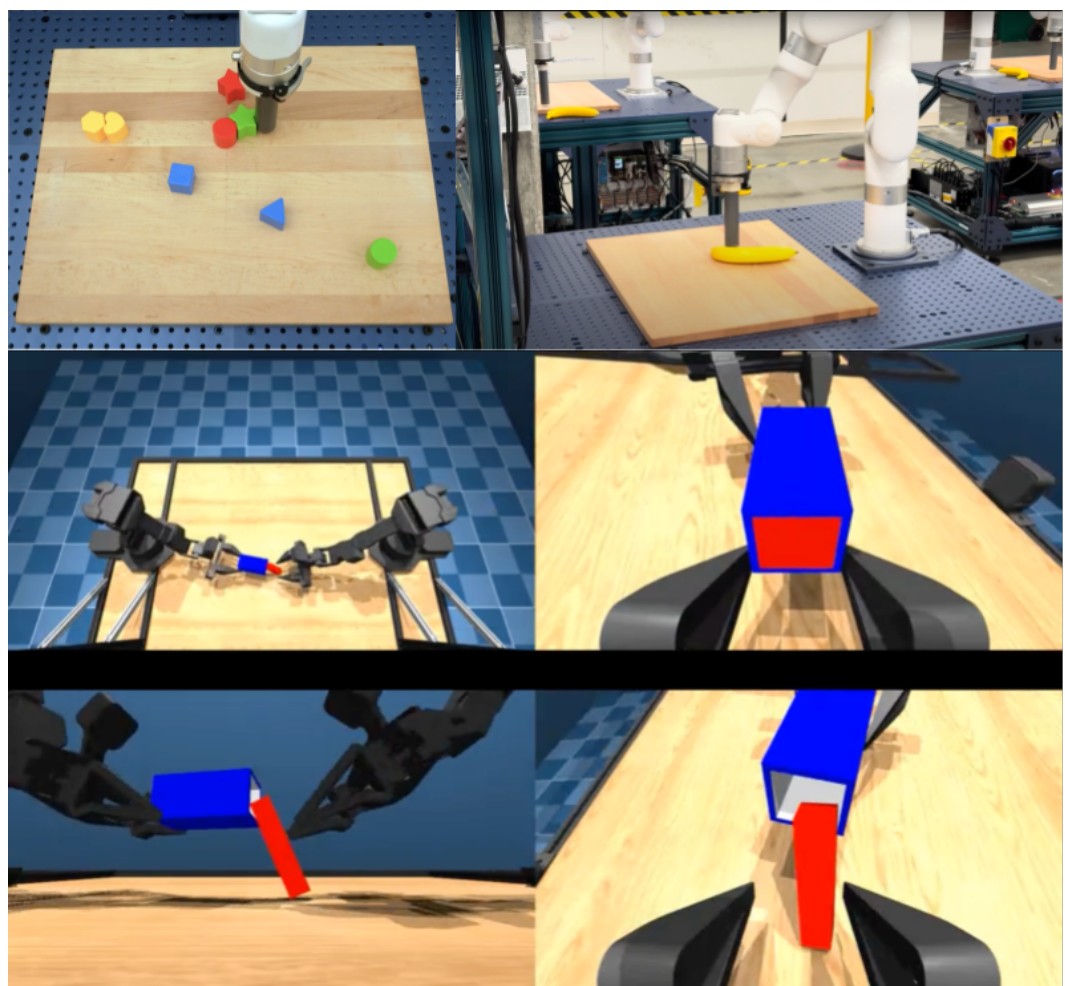

Figure 6: The environments used in this work.

leader robots to control the virtual robots. We then trained a small diffusion policy (Chi et al., 2023) on the 800 demonstrations and used the model to generate 3 datasets of size, 5K, 10K, and 15K.

Critical to successful PaLI policies was to employ semi-global action representations as in Chi et al. (2024), as well as training Stage 1 far beyond the point at which the best validation loss was obtained.

## C  REAL-WORLD LANGUAGETABLE EXPERIMENTATION PROCEDURE

For all real-world experiments, 1 human was responsible for monitoring all robots and performing resets. They did not provide any form of labels or success indicators to the models. Operators were instructed to perform resets either when a block drops off the table, or if a station has not been shuffled and reset in the past 3-5 minutes of operation.

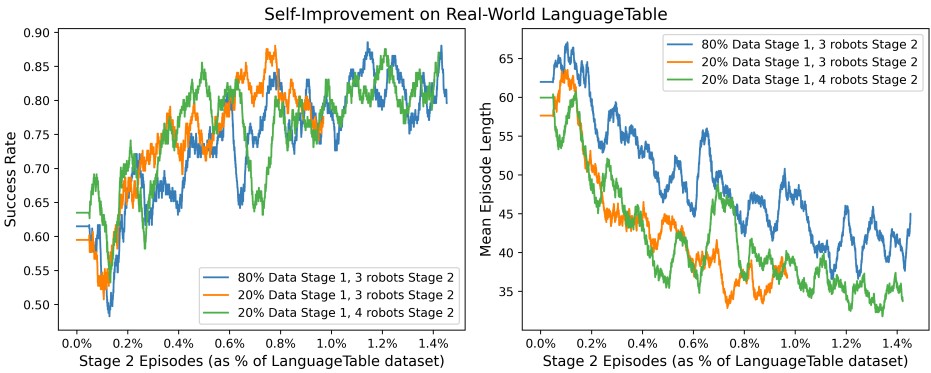

Figure 7: Self-Improvement results on real-world LanguageTable domain. We conducted real-world experiment 3 times: 1) 80% data in Stage 1, Stage 2 fine-tuned on 3 robots simultaneously, 2) 20% data in Stage 1, Stage 2 with 3 robots, 3) 20% data in Stage 1, Stage 2 with 4 robots. In all Stage 2 experiments 1 human monitored and performed period resets for all robots. Each experiment took approximately 20 hours (4 hours × 5 days).

# D    ADDITIONAL PLOTS

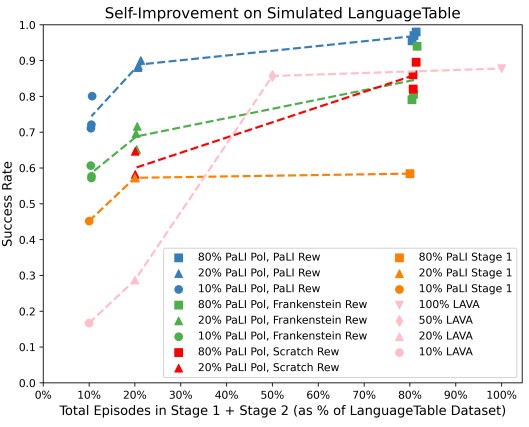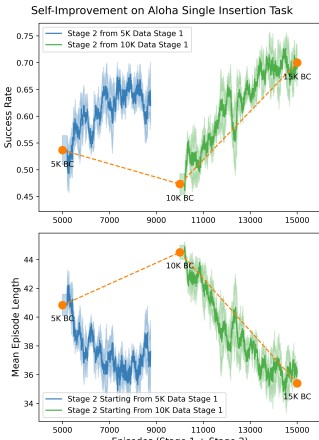

Figure 8: (left) Results and ablations on the simulated LanguageTable domain. We emphasize to the reader that while it appears that the Stage 1 and Stage 2 plots have identical x-axis values, there is no bug in the plot and they are in fact different. The Stage 2 process is simply sample-efficient to the point that the difference in $x$-axis is negligible. (right) Plots demonstrating the efficacy of the Self-Improvement Process on Aloha Single Insertion Task in the 5K and 10K data settings (3 random seeds each setting). The blue plots demonstrate that despite the much smaller datasets compared to LanguageTable, distributing environment interaction budget between Stage 1 and Stage 2 is a more sample-efficient approach towards obtaining performant policies, as opposed to allocating the full budget to Stage 1 (yellow markers).

# E   MATHEMATICAL INTUITION

**Mathematical Intuition**   Let $\mu$ denote the policy corresponding to the imitation learning dataset $\mathcal{D}$ (e.g. the "human policy"). Given the definition in Equation 1, we can see that $d(o_t, g) = -V^\mu(o_t, g)$, where $V^\mu$ denotes the undiscounted value function of policy $\mu$ for the reward function $-\mathbb{1}\left[o_t \text{ satisfies } g\right]$ (i.e. 0 in goal states, -1 elsewhere). Substituting in Equation 2 we obtain, $r(o_t, a_t, o_{t+1}, g) = V^\mu(o_{t+1}, g) - V^\mu(o_t, g)$. Thus, when performing Stage 2 RL updates with discount factor $\gamma$, we have,

$$r(o_t, a_t, o_{t+1}, g) = (1 - \gamma) \cdot V^\mu(o_{t+1}, g) + \underbrace{\left[\gamma \cdot V^\mu(o_{t+1}, g) - V^\mu(o_t, g)\right]}_{\text{reward shaping}} \tag{3}$$

We see that $r(o_t, a_t, o_{t+1}, g)$ is implicitly a shaped reward function (Ng et al., 1999), providing higher rewards in states where the dataset policy $\mu$ performs well. Simplifying the Monte Carlo returns we have,

$$R_t = \sum_{i=t}^{T} \gamma^{i-t} \cdot r(o_i, a_i, o_{i+1}, g) = \left[(1 - \gamma) \cdot \sum_{i=t}^{T} \gamma^{i-t} \cdot V^\mu(o_{i+1}, g)\right] - \underbrace{V^\mu(o_t, g)}_{\text{baseline}}$$

The reward shaping in Equation 3 results in a baseline subtracted from the Monte Carlo returns, leading to lower variance estimates that are particularly useful when employing simple RL objectives, such as REINFORCE in our case. When $\gamma$ is close to 0, we have $R_t = V^\mu(o_{t+1}, g) - V^\mu(o_t, g)$ which is closely similar to a single-step policy improvement for the $-\mathbb{1}[o_t \text{ satisfies } g]$ reward. As $\gamma \to 1$, $R_t$ encourages policies to traverse trajectories along which the states have high value under the dataset policy $\mu$ (i.e. high $V^\mu$). Thus, *performing Stage 2 RL updates with our proposed reward function in Equation 2 leads to policies that more efficiently achieve intended goals while being implicitly regularized to stay close to the dataset policy $\mu$!*

**Toy Pointmass Navigation Domain**   In our supplementary materials website we include an extensive self-contained python notebook implementing our two stage fine-tuning procedure on a pointmass domain. In each episode the pointmass start in a random position, and the goal is for the pointmass to reach a different randomly sampled goal position. We create a purposefully sub-optimal imitation learning dataset for this task, where using a PD-controller we navigate the pointmass to 5 waypoints before heading to the desired goal position. We then execute our proposed Stage 1 and Stage 2 fine-tuning procedures on this imitation dataset using simple MLP policies and steps-to-go prediction models.

Figure 9, depicts the dataset, as well as trajectories from the Stage 1 and Stage 2 policies. Despite the suboptimal behavior of the Stage 1 learned policies, Stage 2 policies clearly converge to an almost optimal policy for the pointmass navigation task. These experiments provide additional support for our choice of reward functions and fine-tuning procedures.

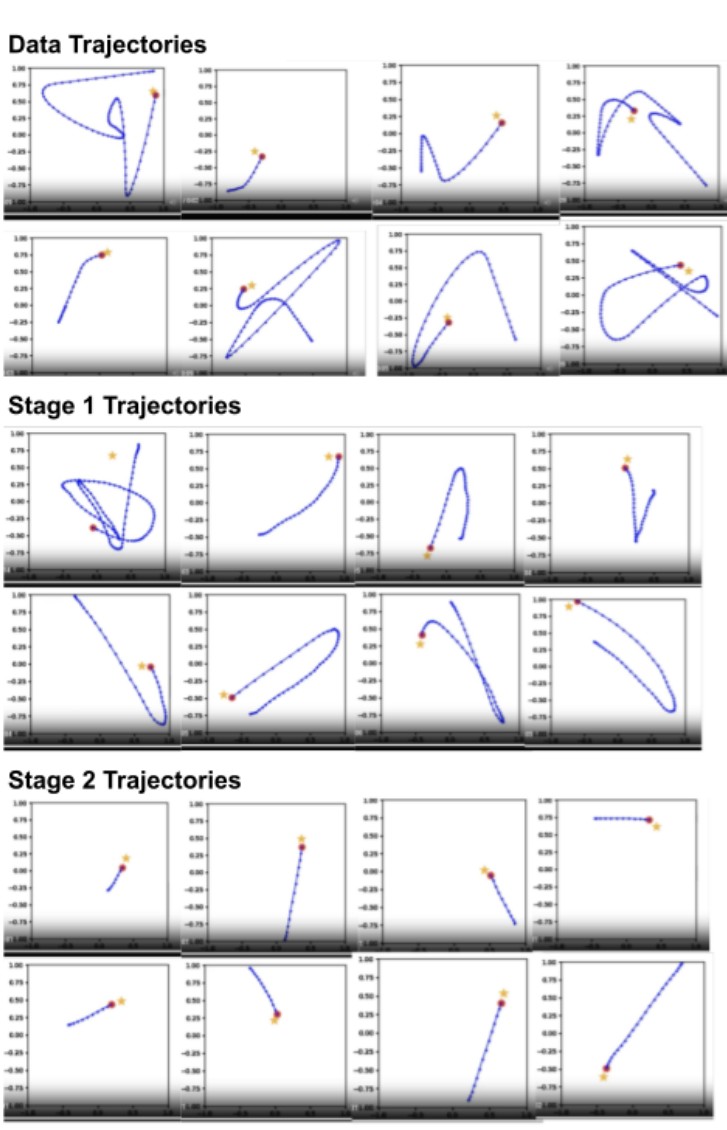

Figure 9: Results from our toy pointmass domain implemented in the python notebook included in our supplementary materials website.

