# OpenReview forum: "On the Surprising Efficacy of Online Self-Improvement for Embodied Multimodal Foundation Models"
_ICLR.cc/2025/Conference — Submitted to ICLR 2025_

### Official Review · Reviewer_5Utp · 2024-10-23

**Soundness:** 3
**Presentation:** 2
**Contribution:** 2
**Rating:** 5
**Confidence:** 4

**Summary:**

This work explores the application of Large Language Model (LLM) training strategies to the web-scale training of a Multimodal Foundation Agent (MFA) for robotics. Specifically, the approach adapts the Supervised Fine-Tuning (SFT) process from LLMs into goal-conditioned behavioral cloning and "steps-to-go" prediction (The model predicts the remaining time steps needed to achieve a given goal) to fine-tune the foundation model. The second phase, analogous to the self-improvement process in LLMs, involves the creation of a reward function. Instead of relying on manual reward design, this method leverages data-driven reward function formation for improved generalization. The reward function is derived from the MFA’s own predictions of the remaining steps-to-go before achieving the goal.

The experiments examine various aspects of implementing the proposed training strategy, including the benefits of the pretraining phase and the improvements gained through the self-improvement stage. Key metrics such as generalization and robustness are analyzed to evaluate the method’s effectiveness.

**Strengths:**

This work focuses on a well-defined setting, utilizing a selected foundation model (PaLI-3B) and a policy architecture similar to RT-2.

The methodology is clearly formalized, with key implementation details explicitly provided. Theoretical insights are also included, offering valuable intuition to aid understanding. Additionally, the use of figures make an attempt to enhance interpretability.

The author conducts extensive experiments across diverse settings (e.g., LanguageTable, ALOHA, BananaTable) to evaluate the model's capabilities. Ablation studies on various components (such as the initialization of the foundation model) are performed to demonstrate the necessity and effectiveness of the design choices.

**Weaknesses:**

A key limitation lies in the use of the time-step variable for the step-to-goal loss, which is central to the pretraining process. However, the time-step is constrained to the range $[0,T]$, which, in my view, limits its generalizability to tasks with arbitrary or longer durations. That said, if the work provides experiments addressing this concern, I am willing to reconsider this point.

Another potential bottleneck arises from the reward function, which remains static due to reliance on frozen pretrained parameters, possibly hindering further performance improvement.

The experiment in Section 5.1 seems insufficient, as it only explores fine-tuning on a narrow subset of tasks. Expanding the study to a broader variety of tasks would enhance its persuasiveness.

Some figures are not clearly presented, particularly Figure 5. I’d appreciate further clarification on its meaning, as I understand it aims to demonstrate the superiority of the BananaTable task, but its message remains unclear.

Certain experiments modify numerous settings without adequately investigating the effect of these changes through ablation studies. For instance, in Section 5.1.3 (the ALOHA experiment), several settings—such as checkpoint selection—are altered, seemingly to push performance boundaries rather than to analyze the impact of individual components. While the system appears promising, a more structured investigation of the contributions of each component would strengthen its case.

The related work section could benefit from deeper analysis of how this work differs from prior studies, rather than merely listing them and leaving readers to infer the differences.

Finally, although this might be a challenging request, the study relies solely on the PaLI-3B model. Evaluating the method on additional Vision-Language Foundation Models would provide more comprehensive insights. Different models adopt varying strategies for input processing and output generation, and these subtle differences could significantly impact fine-tuning results.

**Questions:**

Is there stronger (non-empirical is better) evidence or insight to support why step-to-go prediction is a reasonable approach? My concern is that not all robotic datasets operate under the same speed settings. Even with the same robot, speed may vary across tasks, resulting in significantly different completion times for identical scenarios. This variability could make step-to-go predictions almost random. However, I would consider additional design or ablation studies to address this issue, showing that such speed variations do not have the negative impact I am assuming.

I am concerned that a large improvement in fine-tuning performance might indicate an undesirable outcome. Ideally, the pretrained model should already offer reasonably good performance, and substantial improvement might suggest that the pretraining phase was not as effective as intended. To address this, the reported improvements should be demonstrated through empirical experiments across various tasks, ensuring they reflect multitask generalization or efficient adaption rather than just performance gains on isolated downstream tasks. Could you provide further experiments and insights to validate the observed improvements and confirm that they result from efficient adaptation or general improvements?

---

> ### Comment · Reviewer_Q9jv · 2024-11-25
> **Response to general comments from Reviewer 5Utp**
>
> We thank the reviewer for providing additional related work and insightful questions in their review.
>
> W1. “The experiment section has only a single baseline and the current submission misses several relevant papers [1, 2, 3]. “
>
> We thank the reviewer for bringing attention to these works, which we will include in our related work sections. After reviewing these papers, our response is that these works do not provide fair comparisons because they utilize a-prior information to improve the design-and-control processes. These inductive biases are largely focused on robotic specific settings (symmetrical solutions [2] or access to a robotic design grammar in [3]). The closest promising baseline would be [1]. Our understanding is this algorithm takes a meta-learning approach requiring a distribution of design-and-control tasks to pretrain and adapt the learned design & control policies to new tasks. Our approach trains from scratch which puts our method inherently at a dis-advantage by not utilizing prior design-and-control knowledge to transfer. Combining our work and pre-training could be a promising future work though.
>
> W2 “While the proposed method is novel, the novelty is limited. The idea is closely related to the experience replay idea which is widely used in deep reinforcement learning algorithms.”
>
> Yes, replay buffers are widely used in reinforcement learning algorithms, and  we are the first to consider replay buffers for design generation in joint design-and-control research. This differs from typical replay buffer use which stores transitions of a specific MDP, whereas our replay buffer can be viewed as storing instances of different MDPs in a buffer. This makes the problem notably harder for the controller which is attempting to generalize between different MDPs and our results show the importance of this buffer of MDPs for design-and-optimization.
>
> As another reviewer had a similar comment, we repeat our response above for the reviewer’s convenience below to augment our argument beyond just replay buffers:
>
> We point out that many major contributions in machine learning have been the result of combining existing techniques in the literature. The most relevant to reinforcement learning is the Atari work of Mnih et. al 2013 [1] which combined replay buffers, convolution networks, and target networks to perform complex Atari tasks. Other highly impactful works include Alex Net [2] which applied convolution networks to image net,   as well as Transformers  [3] which combines ideas of attention, batch normalizations and feedforward networks. If desired, we can clarify for each how these works similarly combine existing methods. Our work, similarly, uses established techniques synthesized together to generate superior performance to prior methods that do not combine these methods.
>
> [1] Mnih, V., Kavukcuoglu, K., Silver, D., Graves, A., Antonoglou, I., Wierstra, D., & Riedmiller, M. (2013). Playing Atari with Deep Reinforcement Learning. arXiv [Cs.LG]. Retrieved from http://arxiv.org/abs/1312.5602
>
> [2] Krizhevsky, A., Sutskever, I., & Hinton, G. E. (2017). ImageNet classification with deep convolutional neural networks. Commun. ACM, 60(6), 84–90. doi:10.1145/3065386
>
> [3] Vaswani, A., Shazeer, N., Parmar, N., Uszkoreit, J., Jones, L., Gomez, A. N., … Polosukhin, I. (2017). Attention Is All You Need. CoRR, abs/1706.03762. Retrieved from http://arxiv.org/abs/1706.03762

---

> ### Comment · Reviewer_Q9jv · 2024-11-25
> **Responses to reviewer 5Utp's Questions**
>
> We thank the reviewer for asking several interesting questions which we provide responses to below:
>
> [Q1] In the paragraph titled “Control As A Multi-Step MDP” (around line 216), is it correct to say that the observation space (both that of the environment and that of the design state) can change based on a specific design? If so, how do the authors ensure that a single control policy is compatible with different observation spaces?
>
> We thank the author for this question and we will rewrite this section to reflect this feedback.
>
> To answer the question, Yes, the observation spaces can vary between the designs generated and the generated environment by the design policy.  The control policy π_C operates on a normalized representation of the observation space O_d that maintains consistent dimensionality across designs. The design process corresponds to constructing observation space O_d and action space A_d for the control task, with each design action x_t corresponding to adding or removing subspace tuples (O_i, A_i). This ensures compatibility while maintaining expressiveness.
>
> In practice, we ensure a single policy generalizes to these different spaces by using graph neural networks which can generalize to changing structures of the MDP. We also point out from the reviewer’s suggested baselines that Transformers ([1] in reviewer’s citation list) could be used similarly for this purpose. We do assume that the sequences or graphs have the same continuous vector dimensions (the number of embeddings will change, but not the embedding dimensions themselves).
>
> [Q2] Equation 3 seems incorrect. I think the correct formulation is a nested optimization problem
> d∗=arg⁡maxdJ(πd,d),s.t.πd=arg⁡maxπJ(π,d)
>
> You raise a valid point. While our current formulation captures the high-level objective, your suggested nested optimization formulation more precisely represents the relationship between design and control optimization. The design optimization depends on the optimal control policy for each design. We will update the equation to reflect this nested structure in our revision.
>
> [Q3] In the EDiSon algorithm, the control policy learns from more trajectories as the iteration increases. It is likely that the initial trajectories were poor thus recording lower values for those designs, even if a design is good. This feels like a substantial issue, can the authors comment on this?
>
> The reviewer is correct, earlier designs will not have a fair evaluation early in the design-and-control process. This motivates our use of a replay buffer to provide more accurate estimates of previously considered designs during the joint learning process. Specific to our work, we address incorporate several mechanisms for addressing this in Section 5.2:
>
> - Our design buffer dynamically updates the evaluation of stored designs as the control policy improves
> - The probabilistic storage mechanism p(d) ∝ F(d) naturally adapts to improving performance
> - The bandit-based meta-controller's UCB scoring helps balance between historical performance and potential improvements

---

> ### Author Response · Authors · 2024-11-27
> **Response Part 1**
>
> Thank you for your review of our work. We hope the responses below can address the questions raised:
>
> Weaknesses 1: time-step variable
>
> We can allocate as many tokens as necessary to represent the range [0, T] that covers distribution of episode lengths. With respect to variability in task durations, in tasks considered in the work we already observe quite a wide range of episode lengths. In both the LanguageTable as well as the Aloha imitation learning datasets, the longest episodes can be 4-5x longer than the shortest episodes due to both natural variabilities in initial states, as well as mistakes and retry attempts. As an example, please have a look at the top right video under the "Aloha" section of the supplementary website (https://sites.google.com/view/mfa-self-improvement/home). On each frame of the video, the bottom plot shows the probability distribution of the model’s prediction for steps-to-go at that point in the trajectory. The video demonstrates that our models are able to effectively capture the very multimodal distribution of "steps-to-go", and our experimental results demonstrate that these predictions are effectively utilized in our proposed Stage 2 procedure.
>
> Weaknesses 2: frozen reward
>
> In general the most common setting in reinforcement learning literature is that MDPs are associated with a fixed reward function. Thus we do not believe that this is necessarily a limitation of our work, particularly in light of our strong empirical results as well as the discussion in Appendix E demonstrating that our proposed Stage 2 enables policies to improve beyond the imitation learning dataset. To extend our approach to an iterated setup for future work, we can envision a setting where successful trajectories from Stage 2 are added back to the imitation learning dataset, and the Stage 1 + Stage 2 training steps are repeated.
>
> Weaknesses 3: Section 5.1
>
> During our project, we used careful prompting and LLM APIs to categorize the tasks in the LanguageTable dataset (both sim and real) and found that approximately 47-50% of the tasks in the datasets are varying English descriptions of Block2Block tasks, which are the subset of tasks we considered in LanguageTable Stage 2. Additionally, the remainder of the tasks are closely related as well, many having the form "move the {block name} to the {left/right/top/bottom/etc.} of the table". We would also like to highlight that to ensure fairness, in all of our Figures and data points (e.g. x-axes in Figure 3) we reported the number of episodes based on the number of Block2Block episodes, despite the fact that BC models have access to more training episodes.
>
> Weaknesses 4: Figure 5
>
> Our intention was to give a sense of how much more efficient the BananaTable policies become after Stage 2 fine-tuning. The videos on our supplementary website provide a more clear distinction.
>
> Weaknesses 5: Aloha
>
> When conducting Aloha experiments, we noticed that allowing the Stage 1 BC policies to overfit in terms of BC validation loss resulted in significantly better task success rates. Thus we felt that it may be unfair to the BC baseline to take checkpoints at the optimal validation loss as we had done in the LanguageTable setting. Since our Stage 2 continues from the best Stage 1 checkpoints, we believe the improvement reported in Section 5.1.3 are an accurate reflection of the contribution of our work.

---

> ### Author Response · Authors · 2024-11-27
> **Response Part 2**
>
> Questions 1: Speed Variability
>
> For tasks that can be described as reaching a certain goal state condition (which describes a very broad section of valuable robotics tasks), we believe steps-to-go can be quite reasonable. While steps-to-go can certainly increase the speed at which a robot performs a task, it is not its main contribution as a learning signal. If a policy is not at an expert level when performing a task it will make mistakes along its trajectories, after which it needs to spend extra time recovering from its mistakes and performing retry behaviors. Optimizing for steps-to-go teaches the model to avoid mistake behaviors and be more effective at its task. Indeed our discussion in Appendix E provides support that our approach can lead to policies that outperform the imitation dataset. We do not believe varying speeds across tasks should be a concern. Since the model makes predictions about steps-to-go conditioned on tasks, if a certain task operates at a particular speed the model will learn to predict the correct range values. The other advantage of “steps-to-go” as a signal is that it is a very readily available piece of information. Steps-to-go can also be extracted from non-robotic datasets for pretraining foundation models. E.g. if we have a broad collection of captioned videos (e.g. humans doing arbitrary tasks (https://paperswithcode.com/dataset/something-something-v2, https://ego4d-data.org/)), we can train the model to predict steps-to-go in hopes that it leads to a good initialization for robotic fine-tuning.
>
> Question 2: Large Improvements
>
> In Appendix D, Figure 8, left, we present detailed results in the simulated LanguageTable setting. The orange colored values present success rate after Stage 1 (BC) for our policies, and the pink colored values present success rate for the LAVA policy from the original LanguageTable paper. In both the 10% and 20% dataset regimes, our Stage 1 BC policies obtain 2.5-3x better success rate than the LAVA policies. This gives us confidence that we are effectively training the pretrained foundation models, in particular in the low-data regime which is the focus of our work due to the emphasis on sample-efficiency. In the 80% dataset regime our Stage 1 BC policy performs 1.5x worse than LAVA. Our main hypothesis is that there may be a saturation effect with dataset size: The LAVA model has an almost identical success rate at 50% and 100% data regimes, and our Stage 1 policies have similar success rates at 20% and 80% data regimes. In the Aloha setting, our Stage 1 policies have similar success rates to the policy used to create the imitation learning datasets. Thus, in our presented experiments, the success rate improvements from our proposed Stage 2 are on top of Stage 1 models that already have reasonably good performance.

---

> ### Comment · Reviewer_5Utp · 2024-12-02
>
> Thank you for your response. On a general level, I believe the analysis you provided has addressed some of my concerns and questions, such as the feature of the LanguageTable. However, if your work is published, I would still hope that my concerns—such as time-step and speed—can be addressed through experiments in the final revised version.

---

### Official Review · Reviewer_GL7y · 2024-10-29

**Soundness:** 3
**Presentation:** 2
**Contribution:** 2
**Rating:** 5
**Confidence:** 3

**Summary:**

This paper explores a novel approach to enhance the performance of multimodal foundation models (MFAs) in robotics. The authors propose a two-stage fine-tuning framework:

Stage 1: Supervised Fine-Tuning (SFT), which uses goal-conditioned behavioral cloning and "steps-to-go" prediction objectives.

Stage 2: Online Self-Improvement, where robots practice autonomously using a data-driven reward function based on the "steps-to-go" prediction from the pre-trained model. This eliminates manual reward engineering and enables autonomous practice in simulated and real environments.

**Strengths:**

1. The combination of multimodal foundation agent, supervised fine-tuning, and online self-improvement is a powerful concept that enhances sample efficiency and policy generalization.

2. The method to generate reward functions using expected value of “steps to go” is novel and effective for RL finetuning.

3.  The proposed approach outperforms supervised learning alone and demonstrates generalization capability to new tasks beyond those observed during training.

**Weaknesses:**

1. The method which combines imitation learning with RL finetuning has been used in various areas of robot learning, such as efficient learning [1, 2], bridging human and robot embodiment gap [3], and bridging sim2real gap [4]. This paper just combines this method with Multimodal foundation model, which limits the novelty. Also, adding a paragraph to discuss those related work is helpful.

2. This paper demonstrates that using RL finetuning enhances sample efficiency and policy generalization compare with using only supervised fine-tuning. However, how about only uses pure RL finetuning without supervised fine-tuning? Include an ablation study comparing their two-stage approach to pure RL finetuning without the supervised stage should be helpful.

3. The idea to generate reward functions using expected value of “steps to go” is novel. However, what's the main difference between this idea and using reference trajectory from dataset/ offline policy [2]? The logic might be similar, and it also didn't require human-designed reward. Can you compare the performance of these two methods? Also, there are a lot of related work for generating rewards automatically, such as using human/LLM feedback [5, 6] and metrics functions library [7]. Adding a paragraph to discuss how this paper differs from or improves upon existing methods for automatic reward generation can highlight the contribution.

4. Out of sample efficiency, the another challenge for doing online RL in the real world is that those unreasonable behavior during the exploration process will lead to problems, especially for those  fine-grained tasks and contact-rich manipulation tasks (like the aloha task in this paper). How to avoid these problems? Discuss any safety measures or constraints in the online RL phase in the real world should be helpful.

5. I found the experiment section of this paper paper very difficult to read.

   i. Many of those paragraphs are too long, and a lot of content and images are in the appendix. It is difficult to understand what are the
       tasks at once.

   ii. All the data are placed in paragraphs. This makes the article difficult to read. Perhaps using a table would be a better choice.

[1]. Haldar et al., Watch and Match: Supercharging Imitation with Regularized Optimal Transport, 2022

[2]. Haldar et al., Teach a Robot to FISH: Versatile Imitation from One Minute of Demonstrations, 2023

[3]. Yu et al., MimicTouch: Leveraging Multi-modal Human Tactile Demonstrations for Contact-rich Manipulation, 2024

[4]. Jiang et al., TRANSIC: Sim-to-Real Policy Transfer by Learning from Online Correction, 2024

[5]. Ma et al., Eureka: Human-Level Reward Design via Coding Large Language Models, 2024

[6]. Xie et al., Text2Reward: Reward Shaping with Language Models for Reinforcement Learning, 2024

[7]. Li et al., LEAGUE++: EMPOWERING CONTINUAL ROBOT LEARNING THROUGH GUIDED SKILL ACQUISITION WITH LARGE LANGUAGE MODELS, 2024

**Questions:**

1. How to define the success indicator?

2. The aloha task seems that no semantic information is needed？Why MFA can be benefit?

3. The aloha task requires a large amount of data with additional RL finetuning, but diffusion policy only requires 800 demonstrations. What's the success rate of diffusion policy? Compare with diffusion policy, what are the benefits of using this method?

---

> ### Author Response · Authors · 2024-11-27
> **Response Part 1**
>
> Thank you for your review of our work. We hope the responses below can address the questions raised:
>
> Weaknesses 1:
>
> Thank you for the above references. We will address each group of references separately below:
>
> The works in [1,2] focus on RL-based imitation learning, in a related vein to Adversarial Imitation Learning methods.The key focus of these works is to sample-efficiently imitate the tasks at hand. In contrast, our Stage 2 formulation enables policies to actually improve beyond the trajectories in the imitation dataset. Indeed, both the LanguageTable and the Aloha datasets used in this work contain a very large percentage of suboptimal trajectories, with natural mistakes and recovery behaviors. Each mistake and recovery is costly in terms of steps-to-go progress, and Stage 2 reinforces the model to avoid such behaviors. The Stage 1 and Stage 2 LanguageTable videos on our supplementary website (https://sites.google.com/view/mfa-self-improvement/home) qualitatively demonstrate the more efficient behaviors learned by our Stage 2 policies. For more concrete evidence that our procedure actually results in improvement beyond the behaviors in the imitation learning dataset, we would like to refer you to Figure 9 in Appendix E, which shows results from our supplementary Colab notebook. In the update manuscript on Openreview we have added a paragraph to Appendix E to provide additional details regarding this result. Lastly, compared to [1,2], our approach can generalize to unseen tasks and behaviors, as evidence by BananaTable task in Section 5.3.2.
>
> The work in [3] uses hand-designed reward functions (discussed in their Appendix D), and uses an RL stage to bridge embodiment gaps.
>
> The work in [4] uses hand-designed reward function per task for sim RL training, and subsequently finetunes the model in the real-world using DAgger-style human-in-the-loop fine-tuning. This is an orthogonal problem formulation to our work. In our work the sole role of humans operators is to monitor the robots and periodically reset the robot stations as needed.
>
> Weakness 2:
>
> Tabula Rasa RL without a good policy initialization is generally considered to be sample-inefficient. Since we already assumed access to an imitation learning dataset, we could use this dataset to initialize our policy. Additionally, this allows us to directly compare to one of the most effective foundation based robotics approaches, RT-2, which is equivalent to our Stage 1 policies.
>
> Weaknesses 3:
>
> As discussed in our response above to Weaknesses 1, in contrast to the work in [2], our proposed Stage 2 approach enables 1) policies to improve beyond the behaviors in the imitation learning dataset, 2) policies to practice unseen new tasks (e.g. BananaTable). Additionally, more specifically with respect to the method in [2]: For a broad range of tasks, due to natural variability it can be quite difficult to obtain a large enough reference set of trajectories that is representative of every scenario that may occur. As a simple example, even for a single task in the LanguageTable setting such as "move the red circle to the blue cube", the board can appear in many ways with the blocks and the robot in arbitrary positions. Instead, by teaching the model to predict a progress signal, we can leverage the generalization ability afforded by the underlying foundation model. This effect is highlighted by our ablation experiments (Section 5.2) studying the effect of the foundation model pretraining on the reward model.
>
> Weaknesses 4:
>
> In our current work we do not restrict the exploration of our policy, and do not consider safety effects. Safe exploration is an active area of research and we believe any developments in that setting can be applied to our setting as well. We would also like to highlight some features of our work that may be contribute to safer exploration: 1) Our policies begin the online phase after the BC phase, which should inherently be safer than starting a policy from scratch. 2) Our mathematical intuition section (Appendix E) suggests that our training procedure encourages models to stay close to the dataset distribution, which is a more safe RL update procedure. 3) In our real world experiments, an operator is responsible for monitoring stations, and can intervene in episodes at any time.
>
> Weaknesses 5:
>
> The discussion and datapoints in our paragraphs entirely reference the values in the various figures in our manuscript. We decided to use the paragraphs as an opportunity to reference the key numbers and discuss the conclusions that they are highlighting.
>
> Questions 1:
>
> As discussed in the end of Section 3.2, “Success Detection”, if the steps-to-go predicted by the model (Equation 1) is smaller than a threshold, mark that state as a success state.

---

> ### Author Response · Authors · 2024-11-27
> **Response Part 2**
>
> Question 2:
>
> The pretraining of the underlying vision-language foundation model has already taught the model how to extract and use information from images. Pretrained features have been show to significantly improve robot policy learning (https://arxiv.org/abs/2203.12601, https://arxiv.org/abs/2312.12444).
>
> Question 3:
>
> We used a small diffusion policy in the dataset creation process in order to efficiently train an imitation learning policy with less resources, and use it to rapidly generate larger datasets for our experiments. The success rate of our diffusion policy was approximately 50% which is on the order of our Stage 1 policies. Additionally, we emphasize that our method is not limited to the choice of underlying policy architecture. While the policy architecture we used in this work was derived from RT-2, our approach can for example be directly transferred to the OpenVLA architecture (https://openvla.github.io/) which uses diffusion policies on top of a vision-language foundation model.

---

> ### Comment · Reviewer_GL7y · 2024-11-29
>
> Thanks for providing the response. It has already addressed most of my questions.
>
> Regarding to this current version, I still have some questions and suggestions:
>
> 1. Regarding generating reward functions autonomously, I understand the difference between this approach and matching the expert trajectory now. However, how about the other methods for generating dense rewards? [1,2,3] These reward generation method can generate reward for different tasks and different objec states, which also can generalize to different task settings. I believe more comparison with them should be helpful.
>
> 2. I don't totally agree that do not consider safety effects is doable for this paper since one of the assumption in this paper is that this framework is "reliable and reproducible enough to be employed for real-world robotics". Especially for some more challenging tasks like contact-rich manipulation tasks. Like in the paper [4], the authors shows a complicated method for finetuning offline policy with real-world RL for a contact-rich insertion task, which uses specific reward designs with KL divergence for matching the expert trajectory and resitrict a small actions range to learn residual policy for refine the offline policy. It demonstrates the challenges for doing real world RL. I believe some of the reward designs in this paper (Appendix E) has similar functionality, but add more analysis should be helpful to show that this framework is reliable enough for real-world robotics in more challenging tasks.
>
> [1]. Ma et al., Eureka: Human-Level Reward Design via Coding Large Language Models, 2024
>
> [2]. Xie et al., Text2Reward: Reward Shaping with Language Models for Reinforcement Learning, 2024
>
> [3]. Li et al., LEAGUE++: EMPOWERING CONTINUAL ROBOT LEARNING THROUGH GUIDED SKILL ACQUISITION WITH LARGE LANGUAGE MODELS, 2024
>
> [4]. Yu et al., MimicTouch: Leveraging Multi-modal Human Tactile Demonstrations for Contact-rich Manipulation, 2024

---

> ### Author Response · Authors · 2024-12-04
> **Response**
>
> Thank you for your reply. We hope the responses below address your new comments:
>
> Question 1:
>
> With respect to methods such as [1,2], the iterative process of 1) proposing a reward function, 2) training a policy, 3) refining the reward function and repeating, is untenable for real world robotics. Each real-world training run of a policy with a new reward function requires very significant effort. Furthermore, philosophically, the goal of such approaches is to create a mathematical/symbolic reward function that hopefully results in the intended behavior. In contrast our approach falls under the umbrella of data-driven reward functions, which are more expressive. Consider the task of a humanoid walking to a desired location with a human-like gait. It is incredibly difficult to write down reward functions that capture the intricacies of a natural human gait, whereas data-driven reward functions (such as ours, adversarial inverse RL literature, references from your previous review, etc.) can learn and represent these details directly from the data. Regarding the work in [3], it does not appear that this work is directly related to ours since the focus of their work is to continually build a library of skills, and using the skill library via an LLM planner.
>
> Question 2:
>
> The focus of our work is to demonstrate the efficacy of our proposed self-improvement approach, and the many benefits afforded by the use of foundation models in this process. Safe exploration is an important and very active area of research in the RL and Robot Learning communities, but is orthogonal to the main focus of our work. Nonetheless, safe exploration can certainly be combined with our proposed Stage 2 procedure. Our comment regarding "reliable and reproducible enough to be employed for real-world robotics" refers to the fact that both in real-world and simulation we have shown that our results are reliably reproducible across many trials of the same experiment. In the real-world we conducted 3 LanguageTable experiments (one in the 80% data setting, and two in the 20% data setting) with individual plots in Appendix C Figure 7 left demonstrating reproducibility. In simulation, all our experiments (LanguageTable and Aloha), ablations, as well as Real2Sim experiments were done with many random seeds with minimal variation in results across seeds. The individual markers in Appendix D Figure 8 left, show the results of individual simulated LanguageTable experiments with the blue markers corresponding to our proposed approach. The small blue error bars in Figure 4 left provide another visualization that the variation across experiment seeds is very small. The highlighted regions in Figure 4 right show +/- standard deviation across random seeds which is also small. These extensive results provide confidence in the reliability and reproducibility of our proposed approach.

---

### Official Review · Reviewer_78aV · 2024-11-02

**Soundness:** 4
**Presentation:** 3
**Contribution:** 4
**Rating:** 10
**Confidence:** 3

**Summary:**

This paper employs a method proven effective in training language models to train embodied intelligence models, using reinforcement learning to further refine the model. Following Supervised Fine-Tuning (SFT), the model undergoes an online self-improvement process. In the first phase, the model is trained to clone actions and predict the remaining steps to action success. In the second phase, the predicted remaining steps to task completion serve as a reward for training, enabling reinforcement learning-based optimization. The step count to task completion is also predicted by the model, achieving self-improvement. Validation on the PaLI model showed that the self-improvement procedure increased action success rates by over 1.5 times.

**Strengths:**

This paper introduces reinforcement learning methods into the training of embodied intelligence models, offering a training approach that is more sample-efficient than supervised learning alone. The two-stage training method improves training efficiency, and the second phase enables the model to learn a broader range of actions, enhancing its generalizability. The reinforcement learning section provides detailed explanations of parameter settings. Experiments with the PaLI model validate the practical effectiveness of this training method, supported by real-world robot testing.

**Weaknesses:**

The paper lacks a concluding section, which might impact the overall readability and leave readers without a clear summary of the findings and contributions. Additionally, the arguments supporting the model's generalizability are somewhat limited. The reinforcement learning section would also be strengthened by including more theoretical derivations to better explain the underlying principles.

**Questions:**

1) In the paper, which specific actions are included in the action steps mentioned? Is there a fixed set of actions? When training the model to perform new types of tasks, is it possible to add new actions, or is it limited to the predefined action range?
2) In the middle subfigure of Figure 3, it seems that these exists a more orange box.
3) The authors mentions that the main work is to design an effective and sample-efficient procedure for fine-tuning pretrained multimodal foundation models. Are there any requirements or limitations for multimodal foundation models？What is the impact of the parameter size of the multimodal foundation models on the results, and will the types of multimodal data involved affect the results？
4）In the stage 2, the authors use an online RL to train. Has the online training process considered the safety and coherence of the generated action set?

---

> ### Author Response · Authors · 2024-11-27
> **Response**
>
> Thank you for your review of our work. We hope the responses below can address the questions raised:
>
> Question 1:
>
> As described in our Appendix A, we discretize each continuous action dimension and represent them as tokens in order to train our RT-2 style policies. In the case of LanguageTable, the 2-dimensional continuous actions are converted to a sequence of 4 tokens in the following format: {one token representing + or -}, {one token representing 0-10}, {one token representing + or -}, {one token representing 0-10}. As an example, [-0.13, +0.57] would be tokenized to the token sequence representing -, 1, +, 6. In the Aloha setting we have a 70 dimensional action space. Each dimension’s range is discretized into 256 bins, with one token representing each bin, giving a fine-grained discretization. In general, for any robot embodiment, we can discretize the actions at the granularity that is necessary for that robot, effectively being able to represent any action at high precision.
>
> Question 2:
>
> We additionally trained a Stage 1 (BC) model on a 15K dataset size to provide more datapoints for comparison. As described in the caption, orange colors represent Stage 1 policies, and blue colors represent Stage 2 policies.
>
> Question 3:
>
> The key limitation of foundation models is their size. For choosing the model size, two considerations are important: 1) The model needs to be small enough to enable realtime robot rollouts in the real world. 2) The model needs to be at a size suitable for the training resources that are available. We only performed experiments with the 3B model size as larger models can be too slow to run in realtime on the robot. Intuitively, we would expect that larger models lead to better performance and generalization, as observed in other domains of AI such as NLP. In our work, the ablations in Section 5.2 already demonstrate the key advantage that even our relatively small 3B foundation model provides. With respect to modalities, we believe the foundation models used should be aligned with the robot and task at hand. As an example, having access to audio modality could potentially be valuable for cracking an egg as the audio provides additional signal about the task.
>
> Question 4:
>
> During online training, the policies perform exploration via sampling actions from their distribution which could potentially lead to unsafe actions. Our mathematical intuition in Section 4 (and Appendix E) suggests that our training method will lead to policies that try to stay close to the BC policy, which is favorable in terms of safety characteristics. However, we did not conduct an explicit safety analysis.

---

### Official Review · Reviewer_Q9jv · 2024-11-07

**Soundness:** 3
**Presentation:** 1
**Contribution:** 3
**Rating:** 5
**Confidence:** 3

**Summary:**

The author's propose a novel pipeline to adapt foundation models to robotic low-level control tasks. Their pipeline is a two-stage process. The first is supervised fine-tuning which takes a previously trained foundation model and adapts it to robotic control. Their approach learns to predict the actions and steps to completion during training. Using this pretrained model, they define a reinforcement learning problem using the pre-trained models steps-to-go signal as reward. Including the steps-to-go signal enables their approach not to require a previously defined reward signal, and relies on this prediction as a reward signal. Experimentally,  the author demonstrated that combining a multi-modal pretrained model, their reward function and supervised learning step are necessary to maximize performance.

**Strengths:**

The author’s proposed system doesn’t require a hand-crafted reward signal to do the second fine-tuning stage. It is impressive to see how well this works in their experimental results. The pipeline also seems straightforward to deploy in practice with the understanding that it’s necessary to have large compute available for the foundation models. The authors also conduct several simulation experiments to validate their system and analyze different components in ablation.

**Weaknesses:**

Overall, the paper requires some further work to improve the quality of the manuscript. We leave several suggested edits below. If the authors can address these meaningfully, we believe the paper adequately addresses the target problem (adapting foundation models for low-level control).

Most of our worries with the system are otherwise on the practicality of the approach. The primary issue is that the system requires a second frozen foundation model in the second phase. Storing a second foundation model seems immensely expensive. Consider that some foundation models are at the scales of billions of parameter cases. Perhaps in the current version of the author's framework, this is not an issue. Still, if not in this work, we believe that future work should address compressing this second reward foundation model to make the approach more practical.
Furthermore, real-world robotic experiments need more rigorous evaluation. In the appendix, the authors mention only using a single random seed. Granted, having a single person watch a robot for 20 hours for multiple runs is time-consuming, but this should not be an excuse to properly evaluate the framework's robustness. If this is not feasible, we encourage the authors to discuss this decision, or clearly state better it's a limitation of their evaluation process.

Writing comments:
- Figure 5 is clearly ouside the margins. not adhering to proper margins can be sufficient justification for a desk rejection in some conferences, so please fix it. You can probably cut half the frames for this and keep the intent of the picture
- Should probably use “et al.” in the references for Open X-embodiment [way too many authors]
- Figure 9 in the Appendix is blurry. Either put in clearer pictures or remove them.
- line 042: “Throughout this work…” - sentence feels clunky in the paragraph; rewrite to make a smoother introduction of the term.
- line 087 “our results demonstrate that State 2…” hard to read sentence
- 2 Background -  The author's could move this to the appendix. A better use of background would be to explain the structure of the foundation control “framework” and note that these two models are used (or else push this discussion in methodology as these models are presumably part of the author’s method, whether or not it is unique to these specific models). For example, presumably, you’re using a transformer architecture, which is unclear here. Also, given that you are assuming access to text, formalizing mathematically the structure of your inputs would be more helpful to readers.
- Your algorithm 1 should be referred to as “Algorithm 1” in the main paper (see “Algorithm Box” in line 213)
- Figure 2 - The figure is too big and goes outside the margins of the paper. Reduce the size and improve the quality of the second plot. It could be more vibrant and visually appealing (the y-axis label is too long, the ticks could be bigger, etc.)
- line 249 Mathematical Intuition. Having skimmed Appendix E, it's short enough that you might as well just put it in the main paper or shorten the sentence to “See the appendix for more details”. The current sentence is too long as is.
- Experiments section - I do not like the [Q1, Q2, …, QN] format in the subsections. You could have a preamble in the section saying this section investigates the following Questions instead. It does not look visually appealing as a reader.
- Q1 - Q3, The author's could merge these three questions into a single question
- line 309 “Figure 8 left” the authors should refer to the fact this is in the appendix
- Figure 3 “Values above are averaged across seeds.” How many seeds?
- Figure 3: Author's should add a legend to clarify diagrams
- Figure 3 Why is there a floating orange block in the “Aloha Single-insertion” task?
- Use a more informative name than “Frankenstein.” Some shorthand for what is not included would be more meaningful.
- it would be interesting to see some more meaningful investigation of the claim in Appendix E as it would seemingly give more meaningful results for the second phase of the proposed system.

**Questions:**

- Given the benefits that online training has had in other fields (as discussed in the introduction), what makes the application to robotic control surprising exactly?
- Do you even need these sentence goals in section 3.1? If the prefix is always the same, you could put arbitrary strings for both and achieve the same result, or otherwise not even bother with text.
- line 180 - Why did the authors not add these auxiliary tasks?
- In Equation  1: Is this the actual expectation or a direct prediction of the model? If it’s not clear: \sum p(steps_to_go_i) * steps_to_go_i [ where each option is predicted separately by the foundation model ] vs E(x) = f(o, question) so “expectation” is directly predicted
- Line 214 / Algorithm 1 - Is the small positive constant less than 1? What makes this different from just simply reducing the learning rate?
- Shouldn’t Frankenstein just be called “Pali-ViT without vision-language co-training?”
- What do the authors mean by their results when the Scratch & Frankenstein “underperform” relative to PaLI in Stage 1? Doesn’t including this result by itself support their claim in section 5.2?
-  Figure 7, why do the runs for e.g. 20% State 1, 3 robots end prematurely compared to the other runs?

---

> ### Author Response · Authors · 2024-11-27
> **Response Part 1**
>
> Thank you for your review of our work. We hope the responses below can address the questions raised:
>
> Weaknesses 1: Resources
>
> We would like to argue that once a practitioner has decided to allocate resources for using a foundation model as a policy, our approach Stage 2 process does not necessitate a significant additional investments. If we are running a foundation model policy realtime on a robot, this means that certain compute hardware has been allocated for that robot to perform realtime inference. Therefore, after a rollout, that same hardware resource can be used to load the reward model checkpoint and label the episode. Of course this may lead to lower robot utilization, but it is a tradeoff that can be made if significant compute resource constraints are present. Another interesting avenue for future work would be to explore whether smaller foundation models could be used as effective steps-to-go predictors for reward labelling.
>
> Weaknesses 2: Real World Evaluations
>
> We would like to provide additional clarifications as there may be a misunderstanding: In Section 5.1.2 we mention that "We run the 80% data experiment once using 3 robot stations, and run the 20% data experiment twice, once with 3 and once with 4 robot stations.". Hence in the real world we have conducted 3 full experiments, representing over 60 hours of experimentation time. Appendix C shows the success curves throughout training for each experiment. Additionally, the setup of our real world LanguageTable experiment closely resembles the simulated LanguageTable setup, in which we conducted extensive experiments with many random seeds in every setting and ablation. These two factors provide confidence in our real world results.
>
> Weaknesses 3: Manuscript
>
> We have uploaded an updated manuscript incorporating many of your suggestions. We also include additional discussion in Appendix E regarding the toy pointmass domain, demonstrating that our proposed approach enables policies to improve beyond the imitation learning dataset.
>
> Weaknesses 4: Figure 3 floating orange block in Aloha
>
> We included results for Stage 1 models trained on 15K dataset size to help better situate the Aloha results.
>
> Question 1: Why surprising?
>
> A number of results in our work are surprising for the ML and Robot Learning literatures. 1) The most notable result in our opinion is the sheer sample-efficiency and the significance of the performance boost from our proposed Stage 2, as captured by Figure 3. 2) In contrast to the RL stage of foundation model training in other domains such as NLP, our proposed reward functions are learned from the imitation dataset alone without access to any other forms of ground-truth data, success metrics, or hand-engineered reward functions. 3) Our procedure is able to generalize to very out of distribution scenarios where new behaviors must be learned, e.g. the BananaTable setting. 4) In the current state of Robot Learning research, the use of foundation models is restricted to the behavioral cloning setting. Our results suggest that instead of allocating robot budgets to collecting more imitation learning data, it can be more efficient to enable policies to practice the tasks at hand. This is not intuitively obvious prior to the results of our work.
>
> Question 2: Sentence Goals
>
> The text enables us to specify different tasks, as in the LanguageTable domains. With respect to the prefixes specifically (e.g. “What robot action to”), while they can in theory be replace with a single token representing the query type (i.e. BC vs. steps-to-go prediction), depending on the knowledge embedded in the foundation model from webscale pretraining, the prefixes may help the model better attend to the relevant features in the images. We did not ablate this choice.
>
> Question 3: Auxiliary Tasks
>
> As discussed in lines 181-182, in the LanguageTable setting we did include an auxiliary task where “conditioned on the first and last image of an episode we ask the model to predict what instruction was performed in that episode”.
>
> Question 4: Equation 1
>
> Per our description in Stage 1, the model is trained to maximize the log likelihood of the correct step count token, in the same manner as typical Transformer training. Because tokens are discrete, we can compute the exact expected value in closed form. Thus Equation 1 is computed exactly. In our supplementary materials videos (https://sites.google.com/view/mfa-self-improvement/home), in the first two videos in the Aloha section, the bottom plot displays the probability distribution of the model’s prediction at each step, and the middle plot displays the expectation of the distribution over time.
>
> Question 5: Scaling the loss
>
> Scaling the loss also affects the relative scale of the loss gradients in comparison to regularizer gradients such as weight decay.

---

> ### Author Response · Authors · 2024-11-27
> **Response Part 2**
>
> Question 6: Frankenstein
>
> Indeed, we used “Frankenstein” as a shorthand for "Pali-ViT without vision-language co-training”. We chose this shorthand because when used as a verb "Frankenstein" means: "To combine two or more similar elements into a consistent entity, or a cohesive idea." (https://en.wiktionary.org/wiki/Frankenstein). We felt this was an appropriate characterization since the PaLI model is created by "Frankensteining" a pretrained ViT with a pretrained Transformer to build the initial model checkpoint, and then performing vision-language co-training.
>
> Question 7: Scratch & Frankenstein “underperform”
>
> Despite very extensive efforts, we could not get the Stage 1 policies for Scratch and Frankenstein to obtain non-trivial success rates. Per your comment this already demonstrates the role of foundation model pretraining, as relating to Stage 1. However, it does not provide any color with respect to the role of foundation model pretraining in Stage 2. Our compromise was to use our ablations to study the effect of pretraining on the reward model during Stage 2.
>
> Question 8: 20% 3 robot experiment ending early
>
> At that point in the 3-robot 20% experiment a 4th robot station became available. Since the experiment had already reached a success rate comparable to the best values in the 80% setting, we decided to end the experiment and rerun the 20% experiment with 4 robots. This also allowed us to demonstrate the repeatability of real world experimental results.

---

### Author Response · Authors · 2024-11-27
**Additional Author Comments To All Reviewers**

We would like to thank all reviewers for their detailed comments and valuable feedback, which has already lead to significant improvements to our manuscript. We hope you find that our individual responses to each of your reviews provides additional clarifications. We would also like to take this opportunity to highlight the key aspects of our work that, in our view, make it an important contribution to the field of Robot Learning:

- Our work presents an online self-improvement approach that requires minimal to no modifications to existing multimodal model architectures, and zero access to ground-truth rewards. Furthermore, our approach enables policies to improve beyond the level of demonstrations in the provided imitation learning datasets.
- Our results demonstrate that not only does our online self-improvement process lead to significantly improved policies, but it does so in a manner that is immensely more sample-efficient that relying on supervised learning (Stage 1) alone, despite using on-policy RL with REINFORCE and no data reuse.
- An additional efficiency of our Stage 2 process is that 1 human operator can monitor and reset many robots simultaneously (as done in our real world LanguageTable experiments) since reward labeling and success detection are automatically handled by our framework. This is in stark contrast to imitation learning data collection which requires 1 to 1 human-robot ratio.
- Our extensive experiments demonstrate that our RL approach is incredibly stable and reliable, to the point that we presented results for 4 real world experiment runs, 3 of which were 20 hour long endeavours.
- Critically, we demonstrated 2 generalization experiments that are fundamentally impossible without online training. In particular, our real world BananaTable results (with videos on our supplementary website), demonstrate the unique ability of our approach to not only adapt to novel scenes and semantics, but learn novel behaviors and motions online in order to acquire completely out of distribution skills.

Thank you for your time and engagement in the review process!

---

### Meta-Review · Area_Chair_BaWb · 2024-12-24

**Metareview:**

The paper introduces a two-stage method for using multimodal foundation models in low-level control tasks. In the first stage, the foundation model is fine-tuned to predict actions and "steps-to-go." In the second stage, reinforcement learning (RL) uses the steps-to-go signal as a reward.

Strengths

- The steps-to-go signal serves as a reward, eliminating the need for hand-designed rewards (similar to inverse RL) [GL7y, Q9jv].
- The paper provides a well-defined framework that is easy to use [5Utp, Q9jv].
- The authors conducted several experiments to ablate different components [Q9jv].

Weaknesses

- Real-world experiments in the current version remain limited (e.g., single random seed) [Q9jv, 5Utp] and are not well presented [GL7y].
- The paper does not sufficiently distinguish its contributions from prior works on reward generation and RL fine-tuning [GL7y].

Reasons for Decision

The paper presents a well-executed method for adapting foundation models to robotic control, with notable contributions. However, the limited real-world validation and insufficient analysis of related work diminish its overall impact. These issues indicate that the paper, while promising, requires revision before acceptance.

**Additional Comments On Reviewer Discussion:**

The paper received three weak reject and one strong accept recommendation. While the authors addressed some issues, most reviewers remained unconvinced about the work. We encourage the authors to address these concerns in a revised submission, as the core contributions are compelling and valuable to the field.

---

### Decision · Program_Chairs · 2025-01-22

Reject